# Prediction of spontaneous onset of labor at term (PREDICT study): Research protocol

**Federico Migliorelli**[1,2]*, **Ludovica Ferrero**[1], **Catherine McCarey**[1], **Sara Marcenaro**[1], **Véronique Othenin-Girard**[1], **Antonina Chilin**[1], **Begoña Martinez de Tejada**[1,3]

**1** Division of Obstetrics, Department of Pediatrics, Gynecology and Obstetrics, Geneva University Hospitals, Geneva, Switzerland, **2** Department of Gynecology and Obstetrics, Paule de Viguier Hospital, Centre Hospitalier Universitaire de Toulouse, Toulouse, France, **3** Faculty of Medicine, University of Geneva, Geneva, Switzerland

* femigliorelli@gmail.com

## Abstract

### Background

Recent studies have shown that elective induction of labor versus expectant management after 39 weeks of pregnancy result in lower incidence of perinatal complications, while the proportion of cesarean deliveries remains stable, or even decreases. Still, evidence regarding collateral consequences of the potential increase of induction of labor procedures is still lacking. Also, the results of these studies must be carefully interpreted and thoroughly counter-balanced with women's thoughts and opinions regarding the active management of the last weeks of pregnancy. Therefore, it may be useful to develop a tool that aids in the decision-making process by differentiating women who will spontaneously go into labor from those who will require induction.

### Objective

To develop a predictive model to calculate the probability of spontaneous onset of labor at term.

### Methods

We designed a prospective national multicentric observational study including women enrolled at 39 weeks of gestation, carrying singleton pregnancies. After signing an informed consent form, several clinical, ultrasonographic, biophysical and biochemical variables will be collected by trained staff. If delivery has not occurred at 40 weeks of pregnancy, a second visit and evaluation will be performed. Prenatal care will be continued according to current hospital guidelines. Once recruitment is completed, the information gathered will be used to develop a logistic regression-based predictive model of spontaneous onset of labor between 39 and 41 weeks of gestation. A secondary exploration of the data collected at 40 weeks, as well as a survival analysis regarding time-to-delivery outcomes will also be performed. A total sample of 429 participants is needed for the expected number of events.

**Data Availability Statement:** No datasets were generated or analyzed during the current study. All relevant data from this study will be made available upon study completion, following best practices in

1 / 13

research reporting, by uploading it to Zenodo®
repository, once the results of the investigation
have been published.

**Funding:** The study is financed by the funds of the
Division of Obstetrics at the Department of
Pediatrics, Gynecology, and Obstetrics, of the
Geneva University Hospitals. All the authors are
employees of the Geneva University Hospitals, and
they did not receive specific funding for this work.
The material support (for quantitative Fetal
Fibronectin measurement, qualitative Placental Alfa
Microglobulin 1, and the cervical aspiration device)
was provided free of cost by Hologic, QIAGEN, and
Pregnolia AC, respectively. These founders had no
role in study design, data collection and analysis,
decision to publish, or preparation of the
manuscript.

**Competing interests:** The authors have declared
that no competing interests exist.

## Conclusion

This study aims to develop a model which may help in the decision-making process during follow-up of the last weeks of pregnancy.

## Trial registration

NCT05109247 (clinicaltrials.gov).

## Introduction

Induction of labor (IOL) has long been associated with an increased risk of cesarean deliveries [1, 2]. However, the evidence supporting this observation was based on studies comparing the risk of cesarean deliveries in women who spontaneously started labor with those who required an IOL. Recent studies approached the problem from a different perspective and prospectively compared perinatal outcomes among women who were electively induced at different gestational ages to those who were expectantly managed (including women who enter labor spontaneously, but also women induced later in pregnancy for various indications) [3–5].

The approach of electively inducing labor at certain gestational ages, in order to reduce the incidence of adverse obstetric outcomes (including cesarean deliveries), has been assessed in the settings of advanced maternal age [6, 7], obesity [8], fetal macrosomia [4], or, more frequently, pregnancies over 41 weeks of gestation [3, 9, 10]. Most of these studies concluded that an active attitude towards IOL resulted in lower incidence of perinatal complications, while the proportion of cesarean deliveries remains stable, or even decreases. The most recent example of this strategy has been the "ARRIVE trial" [5], where low-risk nulliparous women were randomly allocated to elective IOL at 39 weeks or to an expectant management group. The main outcome (a composite of neonatal results, including perinatal death) was similar for both groups, but a secondary analysis revealed that the proportion of cesarean deliveries and hypertensive complications was higher in those women assigned to the expectant management group than in the IOL cohort.

Although this recent evidence might support IOL as a method of improving perinatal outcomes, other issues must be addressed. For example, the 0.18% absolute reduction in perinatal mortality after elective IOL at 41 weeks (compared to expectant management) [10] equates to more than 550 inductions needed to avoid one death, while, following the results of the ARRIVE trial, 27 inductions would be necessary in order to avoid one cesarean delivery [5]. Consequently, this strategy would substantially increase the number of inductions and would impact directly on the organization and the resources of contemporary delivery wards.

Additionally, women's opinions in the matter of induction of labor should be considered. In fact, many women feel uncomfortable with the idea of facing an IOL, and more than half of them would rather not repeat the experience in subsequent pregnancies [11, 12]. Therefore, when the benefit is limited, it must be counterbalanced with maternal perceptions, choices, and points of view, especially when IOL will deprive them of the opportunity of spontaneously going into labor.

Having a tool capable of differentiating women who will spontaneously go into labor (thus theoretically benefiting from an expectant management) from those who will require IOL despite any period of expectancy would facilitate individualized counseling and tailored prognosis with regards to their options for managing the last weeks of pregnancy.

The prediction of spontaneous onset of labor becomes the keystone of this approach. However, it is unlikely that a single variable will explain and perfectly classify women into the two groups of interest, as onset of labor is still a poorly known issue and probably related to many processes. Hence, the multifactorial outcome needs to be addressed by a multiple and simultaneous evaluation of many of the features that may be related to the effect of interest, supported by an adequate statistical layout and design.

In summary, our hypothesis is that the combination of different clinical, ultrasonographic, biochemical, and biophysical features will allow prediction of the probability of spontaneous onset of labor during the remaining days of pregnancy, hence allowing discrimination between those women who will spontaneously go into labor from those who will require an induction of labor due to post-term pregnancy.

## Material and methods

### Study overview and aims

We designed a prospective and multicentric observational study including women enrolled at the 39th week of pregnancy, whose deliveries will take place at one of the participating University Hospitals in Switzerland (Geneva University Hospitals and University Hospital Basel). The primary aim is to develop and evaluate a multivariate predictive model able to estimate, between 38+5 and 39+6 weeks of pregnancy, the probability of spontaneous onset of labor before 41 weeks of pregnancy. Secondarily, we will analyze the individual accuracy of each of the evaluated characteristics at 39 and 40 weeks of pregnancy, isolated and within predictive models, to diagnose the onset of spontaneous labor within 1 or 2 weeks. We will also study the influence of each variable at 39 and 40 weeks of pregnancy, alone and in predictive models, in the daily probability of spontaneous onset of labor within 1 or 2 weeks.

### Sample size calculation

According to unpublished data that has been previously collected in our center, approximately 78% of women will start their labor spontaneously before 41 weeks. Hence, considering 10 events (spontaneous onset of labor) for each of the collected variables, 13 participants will be necessary for each of the 15 assessed predictors. For the development of two predictive models (logistic and Cox regressions), these quantities should be doubled which leads to a required sample size of 390 women. An additional 10% has been added to compensate for potential loss of participants, discontinuation due to medical indication of delivery before the post-term period, or the missing values in at least one predictor. Hence, 429 participants will be recruited.

### Eligibility criteria and obtaining informed consent

Term pregnant women who will deliver at any of the participating hospitals will be invited to participate in the study. Inclusion criteria comprise carrying a single and alive fetus, in cephalic presentation and with intact membranes. Gestational age for entering the study should be comprised between 38+5 and 39+6 weeks of pregnancy. Women with fetal malformations, symptomatic uterine contractions, any contraindication for vaginal delivery, medical indication for IOL or elective cesarean delivery, language barrier or inability to give consent, or requesting labor induction will be excluded. Moreover, the use of Pregnolia® aspiration system has specific exclusion criteria, which are listed in the S1 Appendix. If any of these criteria is met, the measurement of cervical stiffness using Pregnolia aspiration device will not be performed, but the participant will still be enrolled.

Patient follow-up will be performed according to local guidelines, without introducing any modification due to their participation. Besides current standard of care, clinical, ultrasonographic biochemical and biophysical parameters will be collected for the purpose of this study. Women will have the possibility to withdraw their consent at any point. Likewise, their participation (or their refusal) will not imply any change in the medical management of the pregnancy.

The recruitment, clinical care and data collection will be organized as follows: The pregnant woman will have her standard medical appointment at 34–38 weeks of gestation and the eligibility criteria will be reviewed. If the criteria are met, the study will be presented to the participant including a study information sheet and a copy of the informed consent. The patient will be contacted by phone within 48 hours to confirm her willing to participate in the study. After confirmation of participation, the first study visit will be scheduled. During this appointment, any study-related question will be answered, and the two copies of the informed consent will be signed (one for the patient and the other to be kept in the study files).

## Outcomes and potential predictors

The main outcome has been defined as a dichotomous variable comprising either the number of participants spontaneously going into labor between 39 and 41+3 weeks of pregnancy or the number of women needing an induction of labor from 41+3 to 42 weeks due to post-term. The secondary outcome was established as the number of days occurring between the date in which the sample collection was performed and the date of delivery. If induction of labor must be performed due to any indication other than gestational age over 41+3 weeks, or if prelabor rupture of membranes occurs and it is not followed by spontaneous onset of labor after 12 hours of expectant management, the record will be censored and only considered for analysis up to the date at which the event took place.

We will collect the following variables as potential predictors of the outcome:

- Clinical variables: maternal age, maternal height, maternal weight at admission, previous vaginal and/or cesarean deliveries (retrieved from medical records), and cervical status, assessed through Bishop score (vaginal examination).

- Ultrasonographic variables: cervical length, fetal head to internal cervical os distance, posterior cervical angle, cervical consistency index (vaginal ultrasound), and estimated fetal weight (abdominal ultrasound).

- Biochemical variables: quantitative fetal fibronectin and qualitative placental alpha microglobulin-1 determinations (obtained from vaginal secretions).

- Biophysical variables: Cervical stiffness index using Pregnolia® system (assessed directly on the cervix using a speculum).

## Study procedures

1. First study visit:

    a. One of the researchers will conduct a directed interview with the pregnant women to collect the required clinical variables, which will be registered in the case report form (CRF).

    b. Woman will be asked to empty her bladder and, once she is ready, to lay on her back in gynecological position.

c. A sterile and disposable speculum will be carefully introduced in the vagina.

d. Two different vaginal samples will be obtained from the posterior fornix using the swabs provided by the manufacturers to quantitatively measure the concentration of fetal fibronectin, as well as to qualitatively evaluate the presence of Placental Alpha Microglobulin-1. Both tests will be performed in situ, with the available commercialized kits for this purpose. The results of these tests will be registered in the CRF, and the vaginal samples and the single-use medical devices will be discarded following hospital guidelines.

e. Next, we will assess cervical stiffness using Pregnolia® system device. The probe must be applied to the anterior cervical lip, if there is no specific contraindication to its use (see S1 Appendix for further information). During the speculum application, three consecutive cervical stiffness measurements (known as Cervical Stiffness Indices) will be obtained and reported in the CRF. The procedure is done according to the instructions for use, considering that cervical stiffness indices are valid only if the first measurement was successful. This exam has been included in the study in November 2021 after an amendment accepted by the ethics committee.

f. The speculum will be retrieved, and a digital vaginal exam will be performed. The five items of Bishop score will be assessed and reported in the CRF.

g. Using a transvaginal sonographic probe, two different images from the uterine cervix will be obtained (ideally on the same frame, using split view):

   i A first sagittal view of the cervix on which the internal and external ora could be identified clearly, obtained without exerting any pressure on the cervix.

   ii A second sagittal view obtained at maximum compression of the cervix, after applying pressure with the transvaginal probe until no further deformation of the tissue is identified, following the previously described technique for cervical consistency index assessment [13].
   Both images will be recorded for off-line analysis. No measurements will be performed at the time of the visit to blind the results from the participant as much as possible.

h. A transabdominal ultrasonographic scan will be performed to assess estimated fetal weight, using Hadlock's formula.

2. If any anomaly is detected during the ultrasound scan (e.g., oligoamnios or fetal malpresentation), the women will be referred to the obstetric unit and managed according to current guidelines.

3. All results will be blinded to physicians in charge and to the participants (double blind).

4. All extracted data will be recorded in the CRF, and the researcher will ask the woman if she is willing to continue with her participation. If so, an appointment will be scheduled 7–10 days later when the woman will be re-examined in the same way, except for fetal weight estimation. Follow-up of the pregnancy will be performed in accordance with current standard of care, with her attending obstetrician or midwife.

5. If labor does not spontaneously commence during follow-up, in the absence of maternal or fetal conditions requiring early delivery, women will be scheduled for induction of labor between 41 and 42 weeks of pregnancy, following local guidelines.

6. A minimal set of data regarding the outcomes of pregnancy will be collected from the clinical file once labor has occurred: type of delivery (vaginal vs. cesarean), indication for

instrumental or cesarean birth, obstetric complications (Hypertension / Preeclampsia, chorioamnionitis, post-partum hemorrhage, 3rd or 4th degree perineal tear, post-partum infections, stillbirth or neonatal death, or others), and neonatal outcomes, such as weight, gender, Apgar scores and umbilical artery pH.

The study pathway for each participant is summarized in Figs 1 and 2. According to our data, we expect that approximately 60% of the participants will remain undelivered for the next consultation. The initial set of data will be collected in all women. However, potential predictors will be again determined in more than half of the participants, resulting in 686 evaluations for a total of 429 women.

The participants may abandon the study at any time upon request. In this case, they will be asked to decide how the information that has already been collected may be used during the analysis, as described in the data management plan.

| | STUDY PERIOD | | | | | | | |
|---|---|---|---|---|---|---|---|---|
| | Enrolment | Allocation | Post-allocation | | | | | Close-out |
| TIMEPOINT | 34-38 w | $38^{5/7} - 39$ w | t = 0 | t + 1w | Dlv. | Dlv. + 1w | etc. | Delivery + 1 month |
| ENROLMENT: | X | | | | | | | |
| Eligibility screen | X | | | | | | | |
| Informed consent explanation | X | | | | | | | |
| Informed consent signature | | X | | | | | | |
| Allocation | | X | | | | | | |
| **INTERVENTIONS:** | | | | | | | | |
| Review of clinical information | | | X | X | | | | |
| Biochemical samples | | | X | X | | | | |
| Measurement of cervical stiffness | | | X | X | | | | |
| Transvaginal ultrasound | | | X | X | | | | |
| Abdominal ultrasound | | | X | X | | | | |
| **ASSESSMENTS** | | | | | | | | |
| Collection of clinical variables | | | X | X | | | | |
| Biochemical analysis | | | X | X | | | | X |
| Collection of perinatal results | | | | | X | | | |
| Analysis of US images | | | | | | | X | |
| Review and closure of record | | | | | | | | X |

Fig 1. Schedule of enrolment, interventions, and assessments. w, weeks; Dlv, delivery; US, ultrasound.

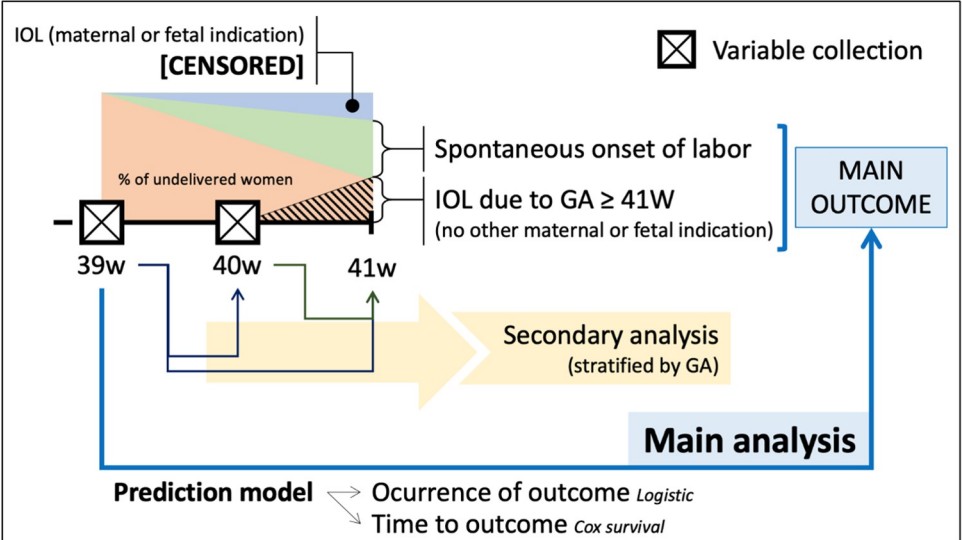

**Fig 2. Study procedures.** Scheme showing the different moments in which the potential predictors will be collected, as well as the outcomes and the planned analysis. A proportion of women will require IOL (or cesarean delivery) due to maternal or fetal indication and will be excluded from the logistic regression models (however considered as censored information for survival analysis). From the remaining cohort, some of them will go into spontaneous labor, while the others will be induced after 41 weeks of pregnancy due to gestational age. These last two groups will define the main outcome. As information will be also collected at 40w, a secondary analysis, stratified by gestational age, will be performed. IOL, induction of labor; w, weeks of pregnancy; GA, gestational age.

## Data management plan

Data will be collected in a form specifically designed for this purpose containing a maximum number of closed questions, adequately codified and anonymized, to avoid the possibility of identification and association of sensitive data to specific participants. All the information will be confidentially handled and will remain digitalized in an informatized database, password-protected and only accessible by the main researchers. Data will be encoded for any kind of transport and all paper forms will be destroyed following current legislation.

All data will be entered to a REDCap® (Nashville, TN) database only by the researchers who were trained in its application, and under utmost confidentiality. One of the researchers -who will not be in charge of data recording- will evaluate data entry by aleatorily selecting 10% of the forms and verifying their correct entry into the database. Also, inadequate or missing values will be periodically assessed to evaluate data quality. This will be periodically performed at all recruiting centers.

If a woman chooses to abandon the study, she will be asked about the use of her information. If the participant refuses to allow the use of the data, the corresponding entrance will be removed from the database and all paper documentation regarding her participation will be destroyed. However, if she agrees to the utilization of data, the follow-up will be censored at the time of consent withdrawal.

In the case that participation should be discontinued due to a medical indication for elective cesarean delivery or induction of labor, censorship will be established from this point. The sample size calculation has been adjusted to compensate for situations of this kind. Regarding loss to follow-up, we consider that delivering outside the Hospital is an unlikely event, as the eligibility criteria specify that women plan to deliver within the participating centers.

Finally, missing data in the predictive variables will lead to the exclusion of the participant from the regression models. Hence, an increase in sample size has been estimated to compensate this issue.

## Statistical analysis

The distribution of continuous or categorical variables will be compared using Student's T test or Fischer's exact test, respectively. Two-sided significance level was set to 0.05.

The main outcome will define two groups: women who spontaneously go into labor and those requiring a post-term IOL. A model of prediction of the binary outcome will be elaborated using stepwise backward and forward logistic regression analysis, including all analyzed features and the gestational age at data collection. To add or remove variables from the model, p-values of 0.05 or 0.10 will be used, respectively. Goodness-of-fit of the model will be evaluated using Hosmer and Lemeshow test and, if multiple models are achieved, the best one will be selected searching parsimony and following Akaike's Information Criterion. Area under the Receiver-Operating Characteristics curve will be used to assess predictive ability of the model. This value will be also calculated using bootstrapping resampling methods (1000 repetitions) to internally validate the results. As only a subset of the sample will have the measurement of cervical stiffness using Pregnolia device (as this assessment was introduced after the start of recruitment), a secondary analysis (following the same principles of the main analysis) will be performed including this variable. A correlation analysis will be carried out comparing cervical consistency index evaluated by ultrasound with cervical stiffness assessed using the aspiration device, as both measurements evaluate the same biophysical characteristic of the cervix.

The probability of spontaneous onset of labor for each day after the analysis will be evaluated using survival regression models. The data will be described using Kaplan-Meier's non-parametrical survival function and the predictive model will be estimated using stepwise backward and forward Cox's regression (proportional hazards model). As mentioned before, data regarding women who have been induced for maternal or fetal indication (different from gestational age) or with prelabor rupture of membrane without spontaneous onset of labor after 12 hours of expectant management will be censored at the time of admission for these causes.

All statistical analysis will be performed using Stata 16.1 software (StataCorp, Texas, USA).

Any incidental finding during analysis will be carefully evaluated and only pondered in the conclusions as hypothesis-generator, considering the multiple biases introduced by the assumption of true findings after studies which were not designed for these purposes.

## Safety considerations

Given that the nature of the study is non-experimental (observational) and that the safety of the interventions has been widely evaluated during pregnancy, women will be not exposed to any risk due to their participation in this research.

This research project will be conducted in accordance with the protocol, the Declaration of Helsinki [14], the principles of Good Clinical Practice, the Human Research Act (HRA) [15] and the Swiss Human Research Ordinance (HRO) [16], as well as other locally relevant regulations.

The project leader will be promptly notified (within 24 hours) if immediate safety and protective measures must be taken during the conduct of the research project. Also, if a serious event occurs, the research project will be interrupted, and the Ethics Committee notified on the circumstances within 7 days according to Swiss HRO.

The participation in the study will represent for women time effort and the need for one extra visit and one extra ultrasonographic exam. However, these have already demonstrated their lack of deleterious effect on the pregnancy, therefore no harm is expected over maternal or fetal health.

No radioactive sources will be used during the development of the research project. Ultrasound examination will be performed following ALARA (As Low As Reasonably Achievable) criteria.

## Status and timeline

Geneva University Hospitals, as the coordinating center, has already started recruitment and about 200 participants have already been included. The two collaborating centers will start including women by the beginning of 2022. If the health situation due to the COVID crisis allows to maintain research activities, we expect to complete the expected sample size by the end of 2022.

## Discussion

We designed this prospective, multicentric, observational study with the aim to identify those clinical, biochemical, biophysical and ultrasonographic factors that may lead to the prediction of spontaneous onset of labor at term.

Considering recent evidence showing no increase in maternal or neonatal morbidity with elective IOL from 39 weeks of pregnancy (even showing an improvement in some indicators, such a cesarean delivery rates) [5], the question regarding the interest of systematically proposing this procedure to all women has been increasingly rising among clinical practitioners in delivery wards. While it may seem reasonable to offer IOL at 39 weeks to women who will end up arriving to the post-term period, so as to reduce complications, it is less intuitive to systematically do so, as it is well-established that women going into labor spontaneously present the lowest cesarean section rates [1, 2]. Also, systematic IOL at 39 weeks would represent an additional medical intervention at term, which may be rejected by many women [11, 12], especially those with low-risk pregnancies.

Unfortunately, it is currently not possible to differentiate women who would benefit from expectant management (and spontaneously go into labor) from those who will end up requiring IOL. Many studies have tackled this idea by searching features or biomarkers able to identify this outcome, mainly evaluating three groups of variables: maternal characteristics (such as age [17, 18], body mass index [17–19], parity [17, 18, 20], ethnicity [17], or Bishop score [21]), ultrasonographic features (including cervical length [17, 20–25] or the presence of funneling [25]), or biochemical biomarkers (comprising the vaginal detection of fetal fibronectin [26, 27]). Most of these studies have been performed in pregnancies arriving at 41 weeks [17, 18, 20, 21], with few carried out at earlier gestational ages [22, 23, 25]. Among all the assessed variables, cervical length and body mass index were the most evaluated features and those that showed a higher association with spontaneous onset of labor [17–25]. Regarding fetal fibronectin, it has mostly been used in the context of preterm labor, and when used at term it failed to discriminate women among the two groups [27]. Placental Alpha Microglobulin-1 has not been used in the prediction of spontaneous term labor.

A biomarker that is gathering interest is the evaluation of cervical consistency. Traditionally included as one of the parameters evaluated in Bishop score, it has never achieved relevance as a predictor neither of preterm delivery nor of spontaneous onset of labor by itself. One of its main flaws is its high subjectivity, which leads to a very low reproducibility. However, different authors have started to focus on cervical consistency as a marker of preterm delivery, assessed either by ultrasound (by means of the cervical consistency index [13]) or using aspiration

devices, as the one developed in Switzerland and commercialized by Pregnolia® AG. Both techniques have shown that cervical consistency decreases with ongoing pregnancy [13, 28], and the ultrasonographic evaluation has also been shown to be a potential predictor of preterm delivery [29], but not of the result of induction of labor [30]). We decided to include the assessment of cervical stiffness with the Pregnolia® AG device later in the study and therefore only a set of women will have this measurement.

We hope this study will allow to integrate current knowledge -along with new collected information- in a single prediction tool to evaluate the chance of spontaneous onset of labor, which will help both pregnant women and their care providers in the decision-making process and in the tailoring of medical advice during the last weeks of pregnancy.

## Ethical considerations

All participants will be adequately informed, orally and written, and the research team will remain at their discretion to answer any question that they may find necessary. Participants will also sign the informed consent form before the study visit and they will have the possibility to withdraw from the study at any point, without any impact on standard of medical care received.

Complications may be diagnosed earlier as each participant will have one extra visit and one additional ultrasound scan. Also, the study does not represent any risk to women, as the explorations that will be carried out are usually performed in current standard of care. All data will remain anonymized, with no risk for participant identification if unauthorized data access occurs.

Regarding the inclusion of vulnerable participants, the recruitment of pregnant women is mandatory for studying the situations involving delivery. The results of this research will help to guide management of pregnant women.

As any work including human subjects, the study has been approved by the Geneva cantonal of the Research Ethics Commission (CCER) (2019–00261).

## Conclusion

This study aims to prospectively develop a model combining relevant clinical, biochemical, biophysical, and/or ultrasonographic variables to predict the spontaneous onset of labor, which will be useful in the decision-making process of the last stages of pregnancy.

## Supporting information

**S1 Checklist. Recommended items to address in a clinical trial protocol and related documents.**
(PDF)

**S1 Appendix. Specific exclusion criteria for Pregnolia® system according to the manufacturer's recommendations.**
(DOCX)

**S1 Protocol. Protocol submitted to ethics committee.**
(PDF)

## Author Contributions

**Conceptualization:** Federico Migliorelli, Catherine McCarey, Véronique Othenin-Girard, Begoña Martinez de Tejada.

**Data curation:** Federico Migliorelli, Ludovica Ferrero, Catherine McCarey, Sara Marcenaro, Véronique Othenin-Girard, Antonina Chilin.

**Formal analysis:** Federico Migliorelli, Begoña Martinez de Tejada.

**Funding acquisition:** Federico Migliorelli, Begoña Martinez de Tejada.

**Investigation:** Federico Migliorelli, Ludovica Ferrero, Catherine McCarey, Véronique Othenin-Girard, Begoña Martinez de Tejada.

**Methodology:** Federico Migliorelli, Véronique Othenin-Girard, Begoña Martinez de Tejada.

**Project administration:** Federico Migliorelli, Catherine McCarey, Véronique Othenin-Girard, Antonina Chilin, Begoña Martinez de Tejada.

**Resources:** Begoña Martinez de Tejada.

**Supervision:** Begoña Martinez de Tejada.

**Validation:** Begoña Martinez de Tejada.

**Writing – original draft:** Federico Migliorelli.

**Writing – review & editing:** Federico Migliorelli, Ludovica Ferrero, Catherine McCarey, Sara Marcenaro, Véronique Othenin-Girard, Antonina Chilin, Begoña Martinez de Tejada.

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
