## [Decision Letter · Decision Letter 0]

13 Apr 2022

PONE-D-21-40209Prediction of spontaneous onset of labor at term (PREDICT Study): Research protocolPLOS ONE

Dear Dr. Migliorelli,

Thank you for submitting your manuscript to PLOS ONE. After careful consideration, we feel that it has merit but does not fully meet PLOS ONE’s publication criteria as it currently stands. Therefore, we invite you to submit a revised version of the manuscript that addresses the points raised during the review process.

 ACADEMIC EDITOR: In agreement with the reviewers ‘comments, the study protocol is in clinically highly relevant topic, and is academically well-designed with clear inclusion and exclusion criteria. Nonetheless, the grammar, flow and details of the manuscript and protocol are not easy to follow, and many points should be clarified and many sentences should be re-wording all along the text.

We look forward to receiving your revised manuscript.

Kind regards,

Guillaume Ducarme, MD, MSc, PhD

Academic Editor

PLOS ONE

Journal Requirements:

4. Thank you for stating the following in the Funding Section of your manuscript:

“The developers of fetal Fibronectin (Hologic®) will provide the assay kits, along with the analyzers, at no cost for the researchers. The supplies to assess the presence of Placental alpha macroglobulin 1 in vaginal secretions will also be offered by the developers (Qiaogen®) without any charge. “Pregnolia AG®” will provide the device for the measurement of cervical stiffness. One Pregnolia Control Unit, as well as the disposable Pregnolia Probes, will be provided free of charge for the conduct of the research.

All the enterprises have been informed that the study protocol will be published, as well as the conclusions of the research, whatever are the results. The design of the study, the data collection, analysis, and publication are an exclusive responsibility of the researchers and no interference from the enterprises is expected. Data will not be shared, and the companies will only know the results once the analysis has finished. All collaboration contracts have been submitted and approved by the Swiss Ethics Committee.

The researchers are workers at the institutions involved in the study and will not receive any compensation for participating in the study. Therefore, the investigators declare no conflict of interest”

We note that you have provided additional information within the Funding Section that is not currently declared in your Funding Statement. Please note that funding information should not appear in the Funding section or other areas of your manuscript. We will only publish funding information present in the Funding Statement section of the online submission form.

“The study is financed by the Geneva University Hospitals. The material required for quantitative Fetal Fibronectin measurement, qualitative Placental Alfa Microglobulin 1 (PartoSure® test), and the cervical aspiration device were provided free of cost by Hologic, QIAGEN, and Pregnolia AC, respectively. These suppliers had no role in study design, data collection and analysis, decision to publish, or preparation of the manuscript. Neither the investigators nor our institution was directly or indirectly financed by these private enterprises. Consequently, the authors declare no conflict of interest.”

Reviewers' comments:

Reviewer's Responses to Questions

**Comments to the Author**

1. Does the manuscript provide a valid rationale for the proposed study, with clearly identified and justified research questions?

Reviewer #1: Yes

Reviewer #2: Partly

Reviewer #3: Yes

2. Is the protocol technically sound and planned in a manner that will lead to a meaningful outcome and allow testing the stated hypotheses?

Reviewer #1: Yes

Reviewer #2: Partly

Reviewer #3: Yes

3. Is the methodology feasible and described in sufficient detail to allow the work to be replicable?

Reviewer #1: Yes

Reviewer #2: No

Reviewer #3: Yes

4. Have the authors described where all data underlying the findings will be made available when the study is complete?

Reviewer #1: Yes

Reviewer #2: Yes

Reviewer #3: Yes

5. Is the manuscript presented in an intelligible fashion and written in standard English?

Reviewer #1: Yes

Reviewer #2: No

Reviewer #3: No

6. Review Comments to the Author

You may also provide optional suggestions and comments to authors that they might find helpful in planning their study.

Reviewer #1: Please revise the following points:

line 57 delete second “labor”, change to “(labor spontaneously)”

line 58 please clarify the meaning of “lately” in this context? Do the authors refer to “recently” or the time point of IOL as in late induction of labor? This is not clear to the reader.

line 60 change “obstetrical” to “obstetric outcomes”

line 72 consider re-wording here “expectant management” (rather than attitude)

line 121 re-wording of heading “Eligibility criteria and obtaining informed consent”

lines 122/123 suggest re-wording to “participate in the study” (rather than join in …)

lines 139/140 suggest re-wording to “If the criteria are met, the study will be presented to the participant including a study information sheet and a copy of the informed consent.”

lines 142/143 suggest re-wording to “the first study visit will be scheduled. During this appointment, any study related questions will be answered and the two copies …”

line 150 “established” (typo)

line 156 maternal height/weight: will BMI be calculated and categorised accordingly?

line 175 rewording “from the posterior fornix”

line 182 change to 2 sentences. : … system device. The probe must be applied …”

lines 204/205 suggest re-wording “ (eg oligohydramnios or fetal malpresentation), change “obstetrical” to “obstetric”

line 210 suggest rewording “later when the woman will be re-examined in the same way, except …”

line 211 follow up of the pregnancy (typo)

line 213 suggest rewording “commence” (rather than start)

line 216 suggest rewording “the study pathway for each participant”

lines 218/219 needs re-wording; suggest “The initial set of data will be collected in all participants. Potential predictors will be again determined in more than half of the women resulting in around 686 evaluations for a total of 429 women.”

line 222 “destination of information”: do you mean storage or release or both? Not clear to the reader. Requires further clarification here.

line 228 suggest re-wording “specific” rather than “concrete”

lines 233/234 suggest re-wording “trained in its application” (rather than handle it), add “be” to change to “not be in charge”, change to “10 % of the forms” (rather than “a 10 % of …”)

line 239 change to “all paper documentation” (delete “the”)

lines 269/270 suggest re-wording “women who have been induced for …” (rather than “whose labors …”)

line 279 do you mean “safety” (not security)?

line 293 needs re-wording “Achievable” not “Acceptable” (ALARA)

line 316 Can you provide a reference for your statement that IOL at 39 weeks is increasing medical intervention at term?

lines 318+319 delete “an”

lines 326-328 this statement needs to be reworded to make it clear to the reader: is it low BMI and short CL which are associated with spontaneous onset of labour?

lines 329/330: suggest re-wording: “Placental alpha Microglobulin-1 has not been used …”

line 337: ad “in” Switzerland

lines 339/340: highlighted/commented statement needs to be re-visited

line 341: meaning of “lately” needs to be defined again here

line 342: suggest re-wording to “current knowledge”

line 349: change to “discretion” (rather than “disposal”), change “They” to “Participants will also sign …”

line 351: suggest re-wording to “ … impact on standard of medical care received”

line 352: suggest re-wording: “Complications might be diagnosed earlier as each participant will have one extra visit and one additional ultrasound scan.

lines 357/358: suggest re-wording: delete “the”, change to “involving delivery. The results of this research will help to guide management of pregnant women” (or something similar)

line 369: do you mean “companies” (enterprises)?

line 370: delete “whatever are the results”

lines 371/372: what do you mean by “interferences”? Needs re-wording. Please change to “companies” again.

line 373: consider re-wording “finalised” (rather than “finished”)

line 375: please change to “employed by the institutions involved in the study”.

The manuscript is written well but needs revision in relation to points raised above. Clinically highly relevant topic. Academically well-designed protocol. Clear inclusion and exclusion criteria.

Reviewer #2: The current manuscript is a research protocol designed to gather data that will then be used to develop a model to determine the prediction of spontaneous onset of labor at term.

The main rationale provided in the paper is that labor induction is associated with increased risk of caesarean delivery, but that the evidence for this was not comparing like for like. The manuscript provides some rationale for the proposed study protocol but would benefit with inclusion and discussion of additional more recently published studies. (please see suggested reference list for inclusion)

Overall, an important topic that could lead to additional information that would help inform management of pregnancy and labor. The authors aim to include biochemical, clinical, ultrasonographic and physiological measurements. The data will then be used to develop a prediction model to determine the participants that will develop spontaneous preterm labor.

The grammar, flow and details of the manuscript and protocol are not easy to follow, and a number of suggestions are included for improvement.

In Lines 77-81, the authors describe the importance of including the opinion of the participant and their perceptions, choices and points of view. But this does not seem to be included within the methods section.

What assessments or questionnaire will be provided to the participant, and how will this data be analysed and included in the final data set, either alone or alongside the biochemical and clinical measurements?

Line 53 – 58: please add citations that support this information.

The manuscript should be revisited and checked for typographical and spelling errors. I have noted a few below.

Line 58: should lately be later? Or other?

Line 101: should ‘would’ be ‘will’?

Line 150: change ‘stablished’ to established

Line 188: what is “XX 2021”

Line 216: “The path gone across by the participant has been summarised in Figure 1” As written this sentence is difficult to understand,

Line 232 – 245: rather than ‘she’ or ‘her’ suggest changing to ‘the participant’ where appropriate. The grammar and flow of this paragraph is difficult to understand.

Line 232: change ‘stablished’ to ‘established’.

Line 360: what is “XXXXXX” Based on this it is not clear if the study has been approved by the ethics committee or not. Please confirm.

The authors track changes and comments seem to still be in places throughout the document.

Additional references to be included and discussed:

Fonseca et al., 2020.

Does induction of labor at term increase the risk of cesarean section in advanced maternal age? A systematic review and meta-analysis.

Eur J Obstet Gynecol Reprod Biol 2020 Oct;253:213-219. DOI: 10.1016/j.ejogrb.2020.08.022

Erickson EN et al 2021

Induction of labor or expectant management? Birth outcomes for nulliparous individuals choosing midwifery care.

Birth. 2021 Dec;48(4):501-513.

Middleton P et al 2020.

Induction of labour at or beyond 37 weeks' gestation.

Cochrane Database Syst Rev. 2020 Jul 15;7(7):CD004945.

Elden H et al 2016

Study protocol of SWEPIS a Swedish multicentre register based randomised controlled trial to compare induction of labour at 41 completed gestational weeks versus expectant management and induction at 42 completed gestational weeks.

BMC Pregnancy Childbirth. 2016 Mar 7;16:49.

Kortekaas JC, et al. 2014.

Effects of induction of labour versus expectant management in women with impending post-term pregnancies: the 41 week - 42 week dilemma.

BMC Pregnancy Childbirth. 2014 Oct 23;14:350.

Stock SJ et al 2012.

Outcomes of elective induction of labour compared with expectant management: population based study.

BMJ. 2012 May 10;344:e2838. doi: 10.1136/bmj.e2838.

Reviewer #3: Interesting abstract for the protocol.

The language could be phrased differently to be clearer, for example (line 52): "Induction of labor (IOL) has longtime been associated with" would read better as "It has been accepted that induction of labour has previously been associated with"

line 93: "will allow to predict" would be better as "will allow prediction of"

line 119" "finally needed" would be better as "recruited".

line 150 and 242: "stablished" - is this "established"?

line 188: "XX 2021" it is now 2022 - is the date known?

line 211: "or" should this be "of"?

line 215: after the patient goes into labour - do you collect the outcome of the labour - i.e. live birth, cesarean required, length of time from hospital arrival to birth or leaving the delivery suite? If you do collect these values, include them in the protocol, and state how they will be reported on.

line 216: "gone across" would be better as "experienced"

7. PLOS authors have the option to publish the peer review history of their article (what does this mean?). If published, this will include your full peer review and any attached files.

Reviewer #1: No

Reviewer #2: No

Reviewer #3: No

---

## [Author Response · Author response to Decision Letter 0]

15 May 2022

Dear Editors and Reviewers, 

You will find below these lines the corrections and observations regarding the revision process. It is my duty to thank you all for your comments and suggestions, which have surely improved our work.

Besides from the recommended amendments, I would like to point out that we have removed the Centre Hospitalier Universitaire Vaudois, in Lausanne, given that the collaboration has not finally fructified. Therefore, the study protocol will only be applied in two Swiss centers.

When pointing out the location of a certain modification in the manuscript, the lines indicated below refer to the ‘Revised manuscript with track changes’ version, so you can easily identify the rectification.

On behalf of all the authors, I would like to show our gratitude once again for the work performed by the Editors and the Reviewers, whose positive and constructive comments have enriched our manuscript. We look forward to receiving your feedback of the corrected version of our work.

We remain at your complete disposal for any further comment or clarification that you may find necessary to address to us

Yours faithfully,

Federico Migliorelli, MD, PhD, 

Corresponding Author

Department of Pediatrics, Gynecology, and Obstetrics, Division of Obstetrics, 

Geneva University Hospitals

30 Boulevard de la Cluse

1211 Geneva 14, Switzerland

E-Mail: femigliorelli@gmail.com

Editors and Reviewers Comments:

Journal Requirements:

Thank you very much for these remarks. We have reviewed the files and updated the file naming, as well as any deviation from your style requirements. We hope that the manuscript will fit your standards in its actual state.

Following your recommendations, we have reviewed the data sharing and availability statements. As mentioned in the cover letter, given that our manuscript describes a research protocol, we are not able to provide any data at this moment, as it has not yet been generated. However, we share your views regarding this issue, and we are committed to publish the minimal underlying dataset to increase reproducibility and transparency once we publish the results of our investigation.

4. Thank you for stating the following in the Funding Section of your manuscript:

“The developers of fetal Fibronectin (Hologic®) will provide the assay kits, along with the analyzers, at no cost for the researchers. The supplies to assess the presence of Placental alpha macroglobulin 1 in vaginal secretions will also be offered by the developers (Qiaogen®) without any charge. “Pregnolia AG®” will provide the device for the measurement of cervical stiffness. One Pregnolia Control Unit, as well as the disposable Pregnolia Probes, will be provided free of charge for the conduct of the research.

All the enterprises have been informed that the study protocol will be published, as well as the conclusions of the research, whatever are the results. The design of the study, the data collection, analysis, and publication are an exclusive responsibility of the researchers and no interference from the enterprises is expected. Data will not be shared, and the companies will only know the results once the analysis has finished. All collaboration contracts have been submitted and approved by the Swiss Ethics Committee.

The researchers are workers at the institutions involved in the study and will not receive any compensation for participating in the study. Therefore, the investigators declare no conflict of interest”

We note that you have provided additional information within the Funding Section that is not currently declared in your Funding Statement. Please note that funding information should not appear in the Funding section or other areas of your manuscript. We will only publish funding information present in the Funding Statement section of the online submission form.

“The study is financed by the Geneva University Hospitals. The material required for quantitative Fetal Fibronectin measurement, qualitative Placental Alfa Microglobulin 1 (PartoSure® test), and the cervical aspiration device were provided free of cost by Hologic, QIAGEN, and Pregnolia AC, respectively. These suppliers had no role in study design, data collection and analysis, decision to publish, or preparation of the manuscript. Neither the investigators nor our institution was directly or indirectly financed by these private enterprises. Consequently, the authors declare no conflict of interest.”

We apologize for the double input regarding the funding information. As asked, we removed the funding-related text from the manuscript, leaving only the Funding Statement on the cover page. Even if the paragraph in the text was much more developed, it essentially contains the same information, so we are keeping the Funding Statement without any further modification. Please let us know if this complies with your standards. We kept on the manuscript a brief sentence disclosing the conflict of interest, which is none for any of the authors.

I believe that there is a misunderstanding regarding the grant information. We did not receive any grant to carry on the research; the study was solely financed by the Department of Obstetrics of Geneva University Hospitals. As stated in the Funding Information, the industry (Hologic, QIAGEN and Pregnolia) supplied the material needed for the study, but neither economic nor financial support was received. I expect that with the abovementioned Funding Statement this issue will be clarified.

The full ethics statement can be found in the lines 418-439 of the manuscript. The letter of approval has been also submitted with the protocol. To clarify this issue, we have introduced the name of the subdivision of the Swiss Ethical Committee approving the protocol, which is the highest authority in this matter. If you believe that further development is necessary, please do not hesitate to ask for it, and we will happily inquire the Committee for any required documentation.

We have added the caption for our Supporting Information at the end of the manuscript (before the references). We have also updated the in-text citations. You can find them in lines 482 and 139, respectively.

Reviewers' comments:

Reviewer's Responses to Questions

Comments to the Author

1. Does the manuscript provide a valid rationale for the proposed study, with clearly identified and justified research questions?

Reviewer #1: Yes

Reviewer #2: Partly

Reviewer #3: Yes

2. Is the protocol technically sound and planned in a manner that will lead to a meaningful outcome and allow testing the stated hypotheses?

Reviewer #1: Yes

Reviewer #2: Partly

Reviewer #3: Yes

3. Is the methodology feasible and described in sufficient detail to allow the work to be replicable?

Reviewer #1: Yes

Reviewer #2: No

Reviewer #3: Yes

4. Have the authors described where all data underlying the findings will be made available when the study is complete?

Reviewer #1: Yes

Reviewer #2: Yes

Reviewer #3: Yes

5. Is the manuscript presented in an intelligible fashion and written in standard English?

Reviewer #1: Yes

Reviewer #2: No

Reviewer #3: No

6. Review Comments to the Author

Reviewer #1: Please revise the following points:

line 57 delete second “labor”, change to “(labor spontaneously)”

line 58 please clarify the meaning of “lately” in this context? Do the authors refer to “recently” or the time point of IOL as in late induction of labor? This is not clear to the reader.

line 60 change “obstetrical” to “obstetric outcomes”

line 72 consider re-wording here “expectant management” (rather than attitude)

line 121 re-wording of heading “Eligibility criteria and obtaining informed consent”

lines 122/123 suggest re-wording to “participate in the study” (rather than join in …)

lines 139/140 suggest re-wording to “If the criteria are met, the study will be presented to the participant including a study information sheet and a copy of the informed consent.”

lines 142/143 suggest re-wording to “the first study visit will be scheduled. During this appointment, any study related questions will be answered and the two copies …”

line 150 “established” (typo)

line 156 maternal height/weight: will BMI be calculated and categorised accordingly?

Indeed, maternal BMI will be calculated. However, we expect to include it as a continuous variable in the model. We may only present the categorization to describe the characteristics of the sample.

line 175 rewording “from the posterior fornix”

line 182 change to 2 sentences. : … system device. The probe must be applied …”

lines 204/205 suggest re-wording “ (eg oligohydramnios or fetal malpresentation), change “obstetrical” to “obstetric”

line 210 suggest rewording “later when the woman will be re-examined in the same way, except …”

line 211 follow up of the pregnancy (typo)

line 213 suggest rewording “commence” (rather than start)

line 216 suggest rewording “the study pathway for each participant”

lines 218/219 needs re-wording; suggest “The initial set of data will be collected in all participants. Potential predictors will be again determined in more than half of the women resulting in around 686 evaluations for a total of 429 women.”

line 222 “destination of information”: do you mean storage or release or both? Not clear to the reader. Requires further clarification here.

line 228 suggest re-wording “specific” rather than “concrete”

lines 233/234 suggest re-wording “trained in its application” (rather than handle it), add “be” to change to “not be in charge”, change to “10 % of the forms” (rather than “a 10 % of …”)

line 239 change to “all paper documentation” (delete “the”)

lines 269/270 suggest re-wording “women who have been induced for …” (rather than “whose labors …”)

line 279 do you mean “safety” (not security)?

line 293 needs re-wording “Achievable” not “Acceptable” (ALARA)

line 316 Can you provide a reference for your statement that IOL at 39 weeks is increasing medical intervention at term?

lines 318+319 delete “an”

lines 326-328 this statement needs to be reworded to make it clear to the reader: is it low BMI and short CL which are associated with spontaneous onset of labour?

lines 329/330: suggest re-wording: “Placental alpha Microglobulin-1 has not been used …”

line 337: ad “in” Switzerland

lines 339/340: highlighted/commented statement needs to be re-visited

line 341: meaning of “lately” needs to be defined again here

line 342: suggest re-wording to “current knowledge”

line 349: change to “discretion” (rather than “disposal”), change “They” to “Participants will also sign …”

line 351: suggest re-wording to “ … impact on standard of medical care received”

line 352: suggest re-wording: “Complications might be diagnosed earlier as each participant will have one extra visit and one additional ultrasound scan.

lines 357/358: suggest re-wording: delete “the”, change to “involving delivery. The results of this research will help to guide management of pregnant women” (or something similar)

line 369: do you mean “companies” (enterprises)?

line 370: delete “whatever are the results”

lines 371/372: what do you mean by “interferences”? Needs re-wording. Please change to “companies” again.

line 373: consider re-wording “finalised” (rather than “finished”)

line 375: please change to “employed by the institutions involved in the study”.

Thank you very much for all your comments and suggestions. We have thoroughly revised the manuscript and corrected the mistakes and sentences as you suggested. We have also rephrased those statements which were not clear for better understanding. We hope that this new version will satisfy the reviewer. 

The manuscript is written well but needs revision in relation to points raised above. Clinically highly relevant topic. Academically well-designed protocol. Clear inclusion and exclusion criteria.

Thank you for your comments. This encourages us to keep on working.

Reviewer #2: The current manuscript is a research protocol designed to gather data that will then be used to develop a model to determine the prediction of spontaneous onset of labor at term.

The main rationale provided in the paper is that labor induction is associated with increased risk of caesarean delivery, but that the evidence for this was not comparing like for like. The manuscript provides some rationale for the proposed study protocol but would benefit with inclusion and discussion of additional more recently published studies. (please see suggested reference list for inclusion)

Thank you very much for your proposal. We have included your suggestions appropriately throughout the paper.

Overall, an important topic that could lead to additional information that would help inform management of pregnancy and labor. The authors aim to include biochemical, clinical, ultrasonographic and physiological measurements. The data will then be used to develop a prediction model to determine the participants that will develop spontaneous preterm labor.

The grammar, flow and details of the manuscript and protocol are not easy to follow, and a number of suggestions are included for improvement.

Thank you very much for your careful reading and your comments. We have corrected the manuscript following your recommendations. We believe that it has been very much improved with your collaboration.

In Lines 77-81, the authors describe the importance of including the opinion of the participant and their perceptions, choices and points of view. But this does not seem to be included within the methods section.

What assessments or questionnaire will be provided to the participant, and how will this data be analysed and included in the final data set, either alone or alongside the biochemical and clinical measurements?

Women’s opinion on matters that affect their pregnancies is always interesting, and that is the reason why we included this description in the rationale for the study. However, this work does not intend to collect information regarding this subject (in fact, we are only collecting variables that may influence on spontaneous onset of labor, but we will not be performing any intervention). In this way, it is out of the scope of this study to assess women’s perceptions of the last weeks of pregnancy. Nevertheless, if we succeed in creating a predictive model of spontaneous onset of labor, its application within a research project and an ulterior clinical protocol could be the matter of an interesting research, focusing on women’s experiences with this tool and the alternatives presented to her to decide on the final stages of pregnancy.

Line 53 – 58: please add citations that support this information.

Thanks for the comment. We thought that this statement was supported by the ulterior references, but we added citations to this paragraph to increase understanding.

The manuscript should be revisited and checked for typographical and spelling errors. I have noted a few below.

Line 58: should lately be later? Or other?

Line 101: should ‘would’ be ‘will’?

Line 150: change ‘stablished’ to established

Line 188: what is “XX 2021”

Line 216: “The path gone across by the participant has been summarised in Figure 1” As written this sentence is difficult to understand,

Line 232 – 245: rather than ‘she’ or ‘her’ suggest changing to ‘the participant’ where appropriate. The grammar and flow of this paragraph is difficult to understand.

Line 232: change ‘stablished’ to ‘established’.

Line 360: what is “XXXXXX” Based on this it is not clear if the study has been approved by the ethics committee or not. Please confirm.

The authors track changes and comments seem to still be in places throughout the document.

Additional references to be included and discussed:

Fonseca et al., 2020. Does induction of labor at term increase the risk of cesarean section in advanced maternal age? A systematic review and meta-analysis. Eur J Obstet Gynecol Reprod Biol 2020 Oct;253:213-219. DOI: 10.1016/j.ejogrb.2020.08.022

Erickson EN et al 2021. Induction of labor or expectant management? Birth outcomes for nulliparous individuals choosing midwifery care. Birth. 2021 Dec;48(4):501-513.

Middleton P et al 2020. Induction of labour at or beyond 37 weeks' gestation. Cochrane Database Syst Rev. 2020 Jul 15;7(7):CD004945.

Elden H et al 2016. Study protocol of SWEPIS a Swedish multicentre register based randomised controlled trial to compare induction of labour at 41 completed gestational weeks versus expectant management and induction at 42 completed gestational weeks. BMC Pregnancy Childbirth. 2016 Mar 7;16:49.

Kortekaas JC, et al. 2014. Effects of induction of labour versus expectant management in women with impending post-term pregnancies: the 41 week - 42 week dilemma. BMC Pregnancy Childbirth. 2014 Oct 23;14:350.

Stock SJ et al 2012. Outcomes of elective induction of labour compared with expectant management: population based study. BMJ. 2012 May 10;344:e2838. doi: 10.1136/bmj.e2838.

Thank you very much for all your comments and contributions. We hope that in its current form, the manuscript is more understandable, and that the methodology section is as clear as necessary to allow replication.

Reviewer #3: Interesting abstract for the protocol.

The language could be phrased differently to be clearer, for example (line 52): "Induction of labor (IOL) has longtime been associated with" would read better as "It has been accepted that induction of labour has previously been associated with"

line 93: "will allow to predict" would be better as "will allow prediction of"

line 119" "finally needed" would be better as "recruited".

line 150 and 242: "stablished" - is this "established"?

line 188: "XX 2021" it is now 2022 - is the date known?

line 211: "or" should this be "of"?

line 215: after the patient goes into labour - do you collect the outcome of the labour - i.e. live birth, cesarean required, length of time from hospital arrival to birth or leaving the delivery suite? If you do collect these values, include them in the protocol, and state how they will be reported on.

Indeed, we will be collecting the labor outcomes once they have occurred, by reviewing clinical files. We have added this information in the protocol, and you may find it in the lines 237-242.

line 216: "gone across" would be better as "experienced"

Thank you very much for all your comments and suggestions. We have included them in our manuscript as we believe that they will increase its quality.

---

## [Editor Report · Decision Letter 1]

23 Jun 2022

Prediction of spontaneous onset of labor at term (PREDICT Study): Research protocol

PONE-D-21-40209R1

Dear Dr. Migliorelli,

We’re pleased to inform you that your manuscript has been judged scientifically suitable for publication and will be formally accepted for publication once it meets all outstanding technical requirements.

Kind regards,

Guillaume Ducarme, MD, MSc, PhD

Academic Editor

PLOS ONE

Additional Editor Comments (optional): All comments and modifications have been addressed
---

## [Editor Report · Acceptance letter]

1 Jul 2022

PONE-D-21-40209R1 

Prediction of spontaneous onset of labor at term (PREDICT Study): Research protocol 

Dear Dr. Migliorelli:

I'm pleased to inform you that your manuscript has been deemed suitable for publication in PLOS ONE. Congratulations! Your manuscript is now with our production department. 

Kind regards, 

on behalf of

Dr. Guillaume Ducarme 

Academic Editor

PLOS ONE